# Regio- and conformational isomerization critical to design of efficient thermally-activated delayed fluorescence emitters

Marc K. Etherington[1], Flavio Franchello[1], Jamie Gibson[2], Thomas Northey[2], Jose Santos[3], Jonathan S. Ward[3], Heather F. Higginbotham[1], Przemyslaw Data[1,4], Aleksandra Kurowska[4], Paloma Lays Dos Santos[1], David R. Graves[1], Andrei S. Batsanov[3], Fernando B. Dias[1], Martin R. Bryce[3], Thomas J. Penfold[2] & Andrew P. Monkman[1]

Regio- and conformational isomerization are fundamental in chemistry, with profound effects upon physical properties, however their role in excited state properties is less developed. Here two regioisomers of bis(10H-phenothiazin-10-yl)dibenzo[b,d]thiophene-S,S-dioxide, a donor–acceptor–donor (D–A–D) thermally-activated delayed fluorescence (TADF) emitter, are studied. 2,8-bis(10H-phenothiazin-10-yl)dibenzo[b,d]thiophene-S,S-dioxide exhibits only one quasi-equatorial conformer on both donor sites, with charge-transfer (CT) emission close to the local triplet state leading to efficient TADF via spin-vibronic coupling. However, 3,7-bis(10H-phenothiazin-10-yl)dibenzo[b,d]thiophene-S,S-dioxide displays both a quasi-equatorial CT state and a higher-energy quasi-axial CT state. No TADF is observed in the quasi-axial CT emission. These two CT states link directly to the two folded conformers of phenothiazine. The presence of the low-lying local triplet state of the axial conformer also means that this quasi-axial CT is an effective loss pathway both photophysically and in devices. Importantly, donors or acceptors with more than one conformer have negative repercussions for TADF in organic light-emitting diodes.

[1] Department of Physics, Durham University, South Road, Durham DH1 3LE, UK. [2] School of Chemistry, Newcastle University, Newcastle upon Tyne NE1 7RU, UK. [3] Department of Chemistry, Durham University, South Road, Durham DH1 3LE, UK. [4] Faculty of Chemistry, Silesian University of Technology, Marcina Strzody 9, 44-100 Gliwice, Poland. Correspondence and requests for materials should be addressed to A.P.M. (email: a.p.monkman@durham.ac.uk).

Thermally activated delayed fluorescence (TADF)[1] has emerged as one of the most attractive methods for achieving luminescence from triplet states via reverse intersystem crossing (rISC)[2] in organic molecules[3,4]. However, designing efficient TADF molecules is not a trivial task due to the restrictions on achieving the correct energy level ordering, splittings and coupling[5]. Until recently, the design focus was based upon an equilibrium picture, where efficient devices were believed to require solely a small singlet-triplet gap. While this energy gap is a critical component, this approach neglects the complexities of the rISC mechanism where subtle effects of molecular properties, such as vibrational degrees of freedom, have recently been shown to be vital[6,7]. Moreover, the roles of molecular geometry and isomeric structures have only recently started to be considered[8,9].

One way of achieving efficient TADF is to use donor–acceptor–donor (D–A–D) molecules possessing strong intra-molecular charge-transfer (CT)[10–12]. However, contrary to initial thoughts[4], rISC is not driven by direct spin–orbit coupling (SOC), which within the one electron limit is forbidden between the [1]CT and [3]CT states. In fact, it is a more complex second-order spin-vibronic SOC mechanism that facilitates efficient rISC. In this process an energetically close local triplet ([3]LE) state acts as a mediator state to couple the [3]CT to the [1]CT states and induce second-order SOC (ref. 6). It is thus the small energy gap between the [3]LE and CT states (both [1]CT and [3]CT) that gives rise to the thermally-activated nature of the rISC[7,13] and efficient OLED performance[14]. The second-order vibrational coupling model of rISC (refs 6,7) enables us to understand the effects of energy ordering, and in this context we show how conformational and regio-isomerism can greatly effect rISC, TADF and ultimately device efficiency. These structural factors must also be considered in designing the highest efficiency TADF emitters along with energy level ordering[13] and host environment within the device[14].

While it is one key factor for TADF that these D–A–D structures exhibit a small $\Delta E_{S_1 - T_1}$ gap, choice of substituents based purely on their donating or accepting properties may still lead to avoidable losses. Phenothiazine (PTZ) is a well-known donor and has been studied in many TADF systems, with great success. However, less well known is the fact that it has two heterogenous conformers that exhibit very different electronic and optical properties[15–17]. Ignoring the potential for conformational changes when using PTZ as a donor molecule may lead to unexpected and unavoidable losses in the system. Lessons learnt from this archetypical donor will apply to all future designs of donor units. Understanding the fundamental limitations imposed by molecular structure and conformation in TADF molecules, which can initially be gauged on their crystal structure, will help avoid those that have such intrinsic loss pathways that inhibit TADF and greatly reduce device efficiency.

There is a great deal of previous work on the observation of dual fluorescence in organic systems[18–22]. However these have overwhelmingly displayed emission from a locally excited (LE) singlet state and a singlet CT state[18–21]. The observation of dual emission arising from two equally stable CT states of a molecule is a more recently discovered phenomenon[22], and in the following systems is as a result of the folding of the PTZ donor unit. It is the intrinsic nature of PTZ, and its ability to form H-intra and H-extra folded conformers that allows formation of parallel quasi-axial (ax) and perpendicular quasi-equatorial (eq) CT states in the bis(10H-phenothiazin-10-yl)dibenzo[b,d] thiophene-S,S-dioxide (DPTZ-DBTO$_2$) molecule. These conformers are taken with respect to the N–S axis and plane of the phenyl rings (nomenclature first suggested by Stockmann et al.[19]). Dual emission from these states has been observed in this work and in the literature[19,20,22].

Here we use two regioisomers of an efficient TADF emitter, bis(10H-phenothiazin-10-yl)dibenzo[b,d]thiophene-S,S-dioxide (DPTZ-DBTO$_2$), to demonstrate that the configuration of the D–A–D molecule has a profound effect on the conformation of the PTZ donor. These conformers are equally stable but based on calculations and optical measurements contribute very differently to triplet harvesting, device efficiency and to the fluorescent properties of these molecules. The two possible conformers of the PTZ result in dual (CT excited state) fluorescence from the molecule, of which only one contributes to TADF and increased device efficiency. The quasi-equatorial conformer is observed in both isomers and yields efficient rISC and TADF. However, the quasi-axial conformer forms a CT state of much higher energy and thus in line with the spin-vibronic coupling mechanism of rISC, this prevents the state from undergoing TADF and contributing to high internal quantum efficiency. Most importantly, the presence of the lower-lying [3]LE state results in the quasi-axial CT state being an effective triplet quencher, which adversely effects device efficiency.

## Results
**Molecular structures**. The two regioisomers have two electron donor units (PTZ) linked to the 2,8- and 3,7-positions on the acceptor unit dibenzothiophene-S,S-dioxide (DBTO$_2$). Figure 1 shows the X-ray crystal structures of 2,8-DPTZ-DBTO$_2$ (2,8-Bis(10H-phenothiazin-10-yl)dibenzo[b,d]thiophene-S,S-dioxide)[11] and 3,7-DPTZ-DBTO$_2$ (3,7-Bis(10H-phenothiazin-10-yl)dibenzo [b,d]thiophene-S,S-dioxide). 2,8-DPTZ-DBTO$_2$ is seen to have both D–A arranged in a quasi-equatorial conformation, whereas 3,7-DPTZ-DBTO$_2$ has a mixed eq–ax conformation. On closer inspection both PTZ units in the 2,8-DPTZ-DBTO$_2$ are in the H-intra conformation, whereas the 3,7-DPTZ-DBTO$_2$ isomer has one H-intra PTZ and one H-extra, of which the H-extra forms the higher-energy quasi-axial CT state. In the case of the H-intra conformation, the nitrogen lone pairs delocalize into the phenyl rings of the PTZ.

**Excited states of each isomer**. Figure 1 shows a schematic of experimental energy levels of the two regioisomers and the effect of polarity. The TDDFT(M062X) calculated dipole moment of the charge-transfer state in 2,8-DPTZ-DBTO$_2$ (15.0 D) is larger than that of 3,7-DPTZ-DBTO$_2$ (14.1 D) consistent for the different shifts of the CT bands as a function of solvent polarity. The HOMO and LUMO levels of the isomers were measured by cyclic voltammetry (Supplementary Fig. 1 and Supplementary Table 1). The LUMO levels of both 2,8-DPTZ-DBTO$_2$ and 3,7-DPTZ-DBTO$_2$ are at the same energy, − 3.05 eV, whereas the HOMO levels are slightly different at − 5.40 eV for 3,7-DPTZ-DBTO$_2$ and − 5.45 eV for 2,8-DPTZ-DBTO$_2$. This narrowing of the HOMO–LUMO gap in 3,7-DPTZ-DBTO$_2$ is also observed in the absorption and emission spectra of the isomers in non-polar methylcyclohexane (MCH) solution, shown in Fig. 2. Careful inspection of the oxidation waves of 3,7-DPTZ-DBTO$_2$ shows a double peak indicative of two species with slightly different oxidation potentials, consistent with the mixed axial-equatorial conformation (Supplementary Fig. 1b). Spectroelectrochemical measurements, along with EPR spectroscopy, reveal the strong decoupling of the D and A in these molecules consistent with their near orthogonality (Supplementary Figs 2–4 and Supplementary Tables 2 and 3).

The absorption spectrum of 2,8-DPTZ-DBTO$_2$ is formed from the sum of the donor and acceptor absorbance, with a weak n–π⋆ absorption band on the red edge that directly creates the [1]CT state[13]. This is consistent with the H-intra conformation where the nitrogen lone-pair electrons are localized on the donor

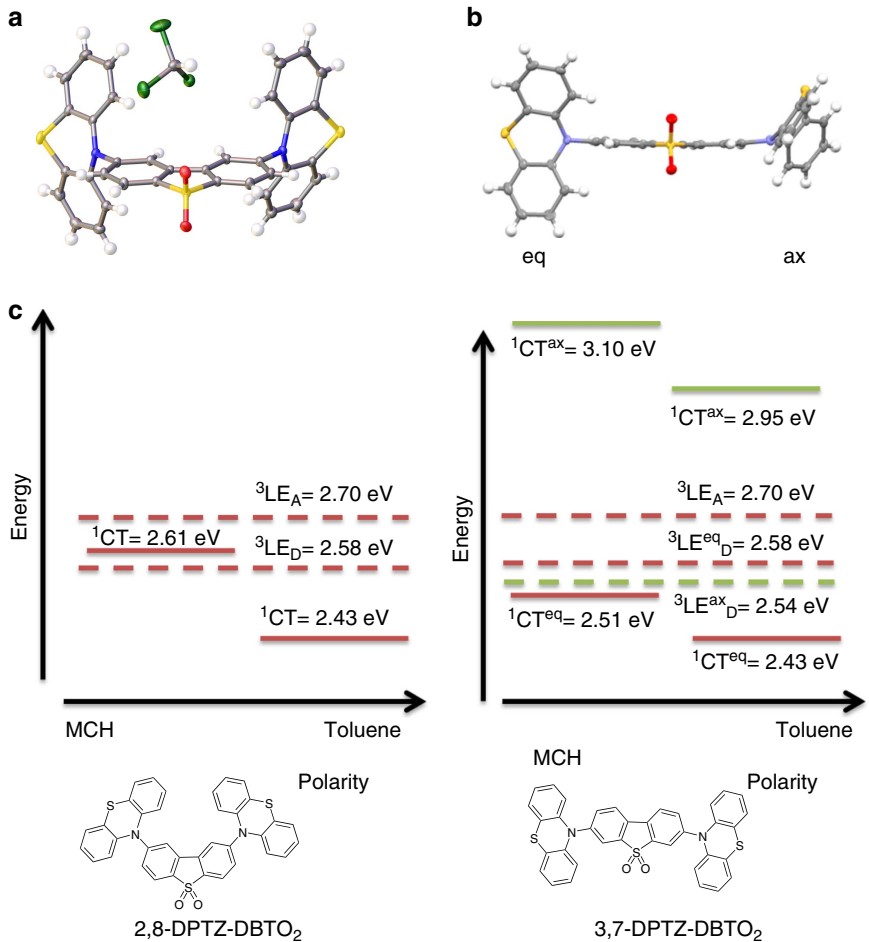

**Figure 1 | X-ray crystal structures and the energy level arrangement of the two isomers.** X-ray crystal structures of the molecules (**a**) 2,8-DPTZ-DBTO$_2$ showing an equatorial PTZ conformation and (**b**) 3,7-DPTZ-DBTO$_2$ having mixed axial and equatorial conformations of PTZ (molecular structure can also be found in Supplementary Fig. 20). (**c**) Energy positions of the $^1$CT and $^3$LE states for the molecules 2,8-DPTZ-DBTO$_2$ (LHS) and 3,7-DPTZ-DBTO$_2$ (RHS). Changing from MCH to toluene for 2,8-DPTZ-DBTO$_2$ moves from type II–III TADF, whereas the equatorial conformer D–A pair (eq)3,7-DPTZ-DBTO$_2$ is type III in MCH and remains so in toluene (see Etherington et al.[7] for notation). Vibronic coupling occurs between the CT manifold and the local triplet excitons (see Supplementary Fig. 12). For the axial (ax) conformer pair the gap is > 0.5 eV and vibronic coupling is suppressed. These values have been extracted from the onset of the $^1$CT emission (Fig. 2b and Supplementary Fig. 6b) and the phosphorescence of the $^3$LE state measured in zeonex (Fig. 6a).

unit. In 3,7-DPTZ-DBTO$_2$ the donor absorption contribution is enhanced and red-shifted, and the lowest energy transitions gain considerable oscillator strength redistributed from the acceptor, suggesting strong mixing of the $n$–$\pi^\star$ and $\pi$–$\pi^\star$ transition moments in 3,7-DPTZ-DBTO$_2$ (ref. 23). This may indicate that the quasi-axial conformer, with its associated H-extra PTZ conformation enhances conjugation between the donor and acceptor via the bridging nitrogen $n$ electrons and hence pronounced mixing of the $n$–$\pi^\star$ and $\pi$–$\pi^\star$ transition moments. These lowest energy absorption bands in 3,7-DPTZ-DBTO$_2$ show no blue-shift with increasing solvent polarity (see Fig. 3b and Supplementary Fig. 5) in contrast to 2,8-DPTZ-DBTO$_2$ (Fig. 3a) where the $n$–$\pi^\star$ states do blue-shift (Supplementary Fig. 6a is not at sufficiently high concentration to observe the blue-shift but is included for completion)[13]. This agrees with TDDFT(M062X) calculations, which yield an oscillator strength of the lowest donor transitions of 0.23 for 2,8-DPTZ-DBTO$_2$ and 0.71 in 3,7-DPTZ-DBTO$_2$, a factor of ca. 3 greater.

Even in a non-polar solvent[24], red-shifted and featureless emission bands are observed in both molecules (see Fig. 2a,b and Supplementary Fig. 6b), indicating the strong CT character of these D–A–D molecules' first excited state. The relative red-shift from the absorption band edge is larger in 2,8-DPTZ-DBTO$_2$

than in 3,7-DPTZ-DBTO$_2$, indicating stronger CT in the former, which can be linked to the H-intra folding of PTZ (found in 2,8-DPTZ-DBTO$_2$) localizing the lone pair of the nitrogen more than the H-extra, giving stronger decoupling of the donor and acceptor. This is confirmed by a smaller overlap of the HOMO and LUMO orbitals on the donor and acceptor groups[25] involved in the CT states for 2,8-DPTZ-DBTO$_2$ (Fig. 4). In 3,7-DPTZ-DBTO$_2$, one sees the HOMO and LUMO are conjugated across the donor and acceptor; which we believe to be mediated by the nitrogen lone pair in the ground state and the *para*-coupling to the equatorial conformer in the excited state. Thus both conformation of the donor–acceptor and the regioisomer configuration play a part in the electronic coupling and mixing of states. Degassing the MCH solutions, leads to a 11-fold increase in the emission from 2,8-DPTZ-DBTO$_2$, but only threefold for 3,7-DPTZ-DBTO$_2$ (Supplementary Figs 7 and 8), indicative of triplet-mediated TADF contributing to the molecular fluorescence, which is greater for the former.

Orbital calculations also show that the mixed ax–eq donor conformation in 3,7-DPTZ-DBTO$_2$ (Figs 1b and 4) results in localization of the HOMO on one of the donor groups, giving rise to two distinct and energetically well-separated CT states (D$_{ax}$–>A and D$_{eq}$–>A) as shown in Fig. 2b. The lower energy

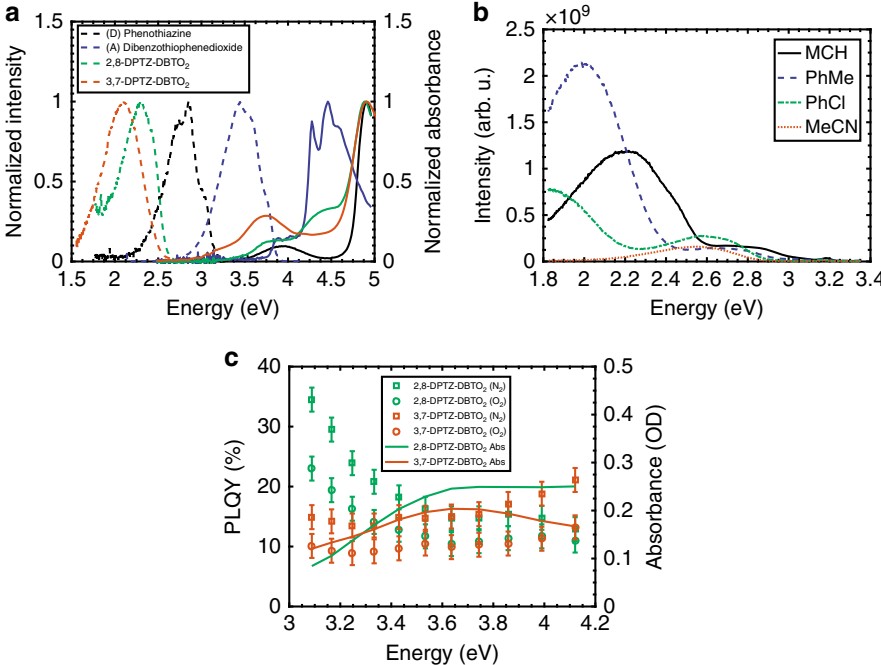

**Figure 2 | Optical properties of the isomers and subunits and solvatochromism of 3,7-DPTZ-DBTO₂.** (**a**) The absorption (solid lines) and emission spectra (dashed lines) of 2,8-DPTZ-DBTO₂, 3,7-DPTZ-DBTO₂ and the subunits in MCH. (**b**) Solvatochromism of the quasi-axial and quasi-equatorial CT states emission in 3,7-DPTZ-DBTO₂. The quasi-equatorial CT shifts strongly as a function of solvent polarity from ∼2.54 eV to below 2.25 eV, however the higher energy quasi-axial CT state displays a much weaker bathochromic shift from 3.1 to 2.9 eV. N.B. The onset for the quasi-equatorial CT appears to be above 2.6 eV in MCH; however, this is the influence of the quasi-axial CT broadening the quasi-equatorial CT spectrum. The method for estimating the CT onsets can be found in the Supplementary Figs 2–6 of our recent work[7]. An example of how this method was used for Fig. 2b is shown in Supplementary Fig. 21, with extracted values in Supplementary Table 15. (**c**) The PLQY of the two isomers with and without oxygen in a zeonex host. The PLQY is shown as a function of excitation energy and the absorption is included to show the wavelength dependency. The error bars are the s.e. based on 10 repetitions for each excitation energy.

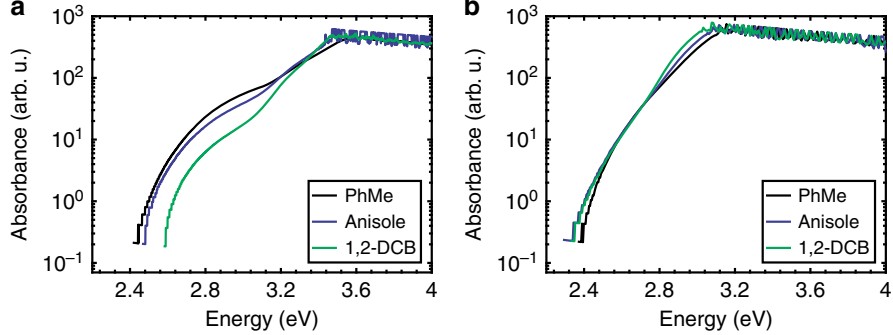

**Figure 3 | High concentration band edge absorption as a function of solvent.** (**a**) The band edge absorption of 2,8-DPTZ-DBTO₂ in a variety of solvents at 1 mM concentration. The hypsochromic shift unveils the $n-\pi^{*}$ nature of the low-energy absorption in this molecule. (**b**) Conversely for 3,7-DPTZ-DBTO₂ at 1 mM concentration there is no significant shift in the absorption thus emphasising the mixed nature of the absorption in this system.

quasi-equatorial CT state ($CT_{eq}$) occurs at 2.54 eV, while the higher energy quasi-axial CT state ($CT_{ax}$) state is at *ca*. 3.1 eV. The two distinct energies of the CT states relate to the electronic properties of the two PTZ conformers, H-intra and H-extra. In the H-extra orientation, which is the origin of $CT_{ax}$, there is less localization of the nitrogen lone-pair into the phenyl rings and a lowering of the HOMO energy. This will lead to a higher-energy CT state as observed, and a weaker CT via the weaker decoupling of donor and acceptor. This arises from the increased conjugation between the donor and acceptor mediated by the lone pairs in the axial conformer. Each regioisomer can, in principle, exhibit three different conformational isomers (ax–ax, eq–eq and ax–eq). The origin for the specific structures of 2,8-DPTZ-DBTO₂ and 3,7-DPTZ-DBTO₂ is energetic, as shown in Supplementary

Table 4, using density functional computations (diagrams of all potential conformers are shown in Supplementary Figs 9 and 10). It is interesting to note that for less polar solvents, such as toluene, this energy gap is reduced and consequently it might be possible for certain conformers to be manipulated by different reaction synthesis conditions.

The two CT states are observed in the solvatochromism of 3,7-DPTZ-DBTO₂ (Fig. 2b), which shows that $CT_{eq}$ is completely quenched in acetonitrile (as in 2,8-DPTZ-DBTO₂), whereas $CT_{ax}$ exhibits weaker CT character, smaller solvatochromic shifts, and is still observable in acetonitrile. This indicates the strong LE character of this axial CT state and relates to the delocalization of donor and acceptor[26]. In zeonex, a rigid non-polar matrix, the absorption spectra are the same as in MCH (Fig. 5) however,

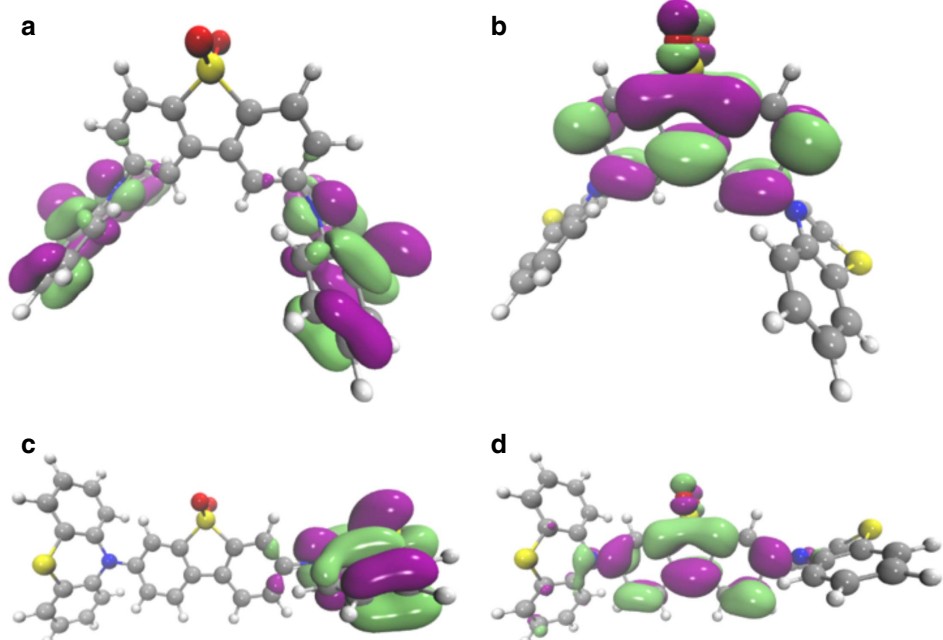

**Figure 4 | Density functional theory calculations of the HOMO and LUMO orbitals on the isomers.** (**a**) HOMO of 2,8-DPTZ-DBTO$_2$ (**b**) LUMO of 2,8-DPTZ-DBTO$_2$ (**c**) HOMO of 3,7-DPTZ-DBTO$_2$ and (**d**) LUMO of 3,7-DPTZ-DBTO$_2$.

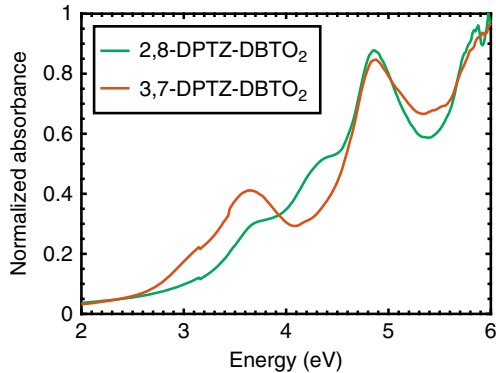

**Figure 5 | The absorption of the two isomers in a zeonex matrix host.** The absorption of 2,8-DPTZ-DBTO$_2$ and 3,7-DPTZ-DBTO$_2$ in a zeonex matrix host, which are the same as their absorption in MCH solution. Again highlighting that the donor absorption contribution is enhanced and red-shifted, and the lowest energy transitions gain considerable oscillator strength redistributed from the acceptor, suggesting strong mixing of the $n$–$\pi^\star$ and $\pi$–$\pi^\star$ transition moments in 3,7-DPTZ-DBTO$_2$.

degassing reveals that a significant fraction of increased intensity in the emission comes from $^3$LE phosphorescence (see Supplementary Fig. 11) in 3,7-DPTZ-DBTO$_2$. To understand these differences and the effect of $n$–$\pi^\star$ and $\pi$–$\pi^\star$ mixing, the photoluminescence quantum yield (PLQY) was measured as a function of excitation energy and oxygen content, Fig. 2c.

Both regioisomers show higher PLQY in inert atmosphere as expected. 2,8-DPTZ-DBTO$_2$ exhibits an increase in the PLQY ($>$40%) around the absorption band edge ($\sim$3 eV) due to the direct excitation of the $^1$CT states via the $n$–$\pi^\star$ transition[13]. In 2,8-DPTZ-DBTO$_2$ this PLQY increase is also seen in the absence of oxygen, indicating competition between $^3$LE$_D$ and $^1$CT formation, a result of slow electron transfer[13]; however, the $^3$LE$_D$ states formed from the $^1$LE$_D$ are still harvested by rISC. The effect on PLQY clearly shows that the $^1$LE$_D$ ISC is a very efficient

quenching channel for the excited donor. The general increase in PLQY is also observed in 3,7-DPTZ-DBTO$_2$ (when compared to the decreasing absorption cross section) but there is no sharp increase at the band edge. This is due to the strongly mixed $n$–$\pi^\star$ and $\pi$–$\pi^\star$ character of its low-lying transitions preventing direct $^1$CT formation. The radiative decay of $^1$LE$_D$ competes with electron transfer and ISC, which will reduce the overall yield of delayed emission[13].

The phosphorescence spectra of 2,8-DPTZ-DBTO$_2$ and 3,7-DPTZ-DBTO$_2$ in a zeonex matrix are shown in Fig. 6a. 2,8-DPTZ-DBTO$_2$ shows contributions from both $^3$LE$_A$ (2.70 eV) and $^3$LE$_D$ (2.58 eV) phosphorescence as previously observed[13], whereas the band shape in 3,7-DPTZ-DBTO$_2$ exhibits less well-resolved vibronic components. From the phosphorescence onset, the energy of the lowest triplet state is 2.54 eV. This is below that of the $^3$LE$_D$ in 2,8-DPTZ-DBTO$_2$, and corresponds to the local donor triplet state of the H-extra PTZ, $^3$LE$_{D,ax}$. Critically, we note that the CT state onsets of this conformer are at *ca.* 3.1 eV (Fig. 2b), consequently the CT-$^3$LE gap is much larger than that of the equatorial conformer, by *ca.* 0.5 eV and this gap is sufficiently large to make rISC unlikely, therefore only $^3$LE$_{D,ax}$ phosphorescence should be observed from this axial conformer. Energy transfer from $^3$LE$_{D,eq}$ to $^3$LE$_{D,ax}$ is improbable given the donors orthogonality and large spatial separation. This observation now helps to explain why in certain sterically hindered D–A–D systems where both PTZ donors are stabilized in the H-extra conformer, no delayed $^1$CT fluorescence (DF) at all is observed, only very strong room temperature phosphorescence, because the S-T gap is too large and the energetically low-lying axial triplet acts as a sink for all excitations[27].

**Photoinduced absorption.** To confirm the presence of a $^3$CT$_{eq}$ population, which is the lowest energy triplet state of 3,7-DPTZ-DBTO$_2$, photoinduced absorption (PIA)[28] of the isomers in zeonex was measured (Fig. 6b). Both isomers show a characteristic, slightly structured induced absorption in the region of the $^3$LE$_{D,eq}$ T$_1 \rightarrow$ T$_N$ absorption[29–31]. This absorption is more

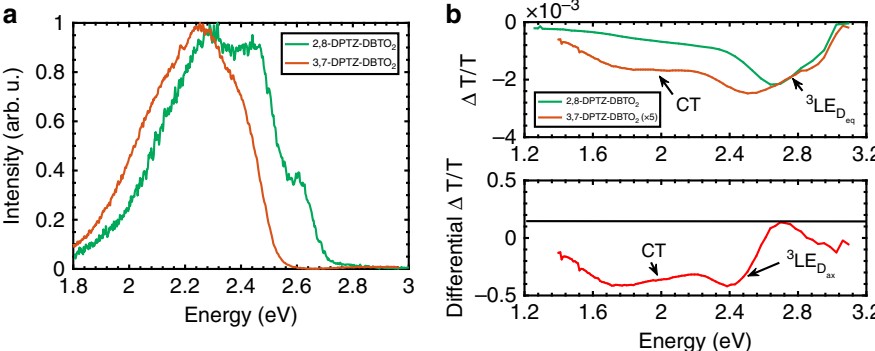

**Figure 6 | Phosphorescence emission and the out of phase and differential PIA of the DPTZ-DBTO$_2$ isomers. (a)** The phosphorescence spectra are measured at 80 K in a zeonex matrix at 0.5 ms delay time. **(b)** The peak that is located below 2.4 eV in both spectra is attributed to the T$_1$–T$_N$ absorption of phenothiazine. The broad absorption up to 1.4 eV found in 3,7-DPTZ-DBTO$_2$ is attributed to the CT states emphasizing the energy diagram shown in Fig. 1c. The normalized differential is shown to emphasize the CT state PIA and the shift in the $^3$LE PIA.

defined in 2,8-DPTZ-DBTO$_2$, and is consistent with population of this state by both directly photocreated $^3$LE$_D$ states (ISC from $^1$LE$_D$) and those created via efficient ISC from $^1$CT. In contrast, 3,7-DPTZ-DBTO$_2$ has an additional broad induced absorption between 2.2 and 1.4 eV, which is attributed to the $^3$CT$_{eq}$ state[7], and has characteristics similar to the donor cation[19,20], (Supplementary Fig. 2), consistent with the $^3$CT$_{eq}$ state being the lowest energy state, and the observed DF lifetime of ca. 13.7 μs. The signal in 3,7-DPTZ-DBTO$_2$ is also 5 times smaller than in 2,8-DPTZ-DBTO$_2$, consistent again with the larger pool of triplet excitations initially formed in 2,8-DPTZ-DBTO$_2$, but there is also a contribution in the 3,7-DPTZ-DBTO$_2$ spectrum from this direct $^3$LE$_D$ production channel.

Subtracting the normalized PIA signals of 2,8-DPTZ-DBTO$_2$ from 3,7-DPTZ-DBTO$_2$ the $^3$CT$_{eq}$ induced absorption in the 3,7-DPTZ-DBTO$_2$ signal is elucidated and also a low-energy triplet PIA component, $\sim$200 meV below that of the $^3$LE$_{D,eq}$ T$_1 \rightarrow$ T$_N$ transition, ascribed to the $^3$LE$_{D,ax}$ T$_1 \rightarrow$ T$_N$ transition. This supports that $^3$LE$_{D,ax}$ is lower in energy than $^3$LE$_{D,eq}$ in 3,7-DPTZ-DBTO$_2$.

**Time-resolved spectroscopy.** Figure 7 shows the emission decay profiles of the two isomers in degassed solutions (MCH and toluene; see Supplementary Table 5 for parameters). In all decays, there are two characteristic time regimes; the first (early times) is related to prompt $^1$CT fluorescence (PF); the second (longer times) relates to the DF. This DF arises from excitations harvested from the triplet states ($^3$LE) via rISC, as it is quenched by oxygen. The spectra of the prompt and delayed CT emission correspond exactly in both 2,8-DPTZ-DBTO$_2$ and 3,7-DPTZ-DBTO$_2$, and the DF has linear power dependency in both cases as expected for TADF (ref. 32; Supplementary Figs 13 and 14).

In 3,7-DPTZ-DBTO$_2$, we observe fast emission from CT$_{ax}$ appearing in the first couple of nanoseconds, however, the measurement system has insufficient time resolution to resolve it. In MCH the PF lifetime of 2,8-DPTZ-DBTO$_2$ is twice that measured in 3,7-DPTZ-DBTO$_2$, 15.4 ± 0.9 ns compared to 7.7 ± 0.3 ns, in line with the calculated oscillator strengths of the $^1$CT states of both isomers, $2.3 \times 10^{-4}$ for 2,8-DPTZ-DBTO$_2$ and $5.1 \times 10^{-4}$ for 3,7-DPTZ-DBTO$_2$. This reflects the $\pi^\star$–$n$ and $\pi^\star$–$\pi$ coupling of each to the ground state. The DF in 2,8-DPTZ-DBTO$_2$ is 15 times stronger than 3,7-DPTZ-DBTO$_2$ (Fig. 7). This disparity is because of the enhanced population of $^3$LE states formed by direct ISC from the long-lived $^1$LE state in 2,8-DPTZ-DBTO$_2$, compounded by excited states lost via the axial conformer channel in 3,7-DPTZ-DBTO$_2$. This is a peculiarity

of optical excitation and is different to the mechanisms that will occur in a device. The DF lifetime in 3,7-DPTZ-DBTO$_2$ is found to be 3.8 ± 0.2 μs and for 2,8-DPTZ-DBTO$_2$ 5.2 ± 0.3 μs. The DF rate is a combination of the rISC rate and the $^1$CT radiative rate and the difference between 2,8-DPTZ-DBTO$_2$ and 3,7-DPTZ-DBTO$_2$ is a manifestation of the small oscillator strength of the former and the rISC rate of the D$_{eq}$-A unit of 3,7-DPTZ-DBTO$_2$ being half that of 2,8-DPTZ-DBTO$_2$ (ref. 9; Fig. 8, with parameters shown in Supplementary Tables 6–8).

A factor of two increase in the prompt $^1$CT emission lifetime occurs when the molecules are in a more polar solvent (toluene). This increase is related to higher CT stabilization through the polar medium (Fig. 1c), which reduces the energy of the $^1$CT state and hence increases the energy gap between the CT and $^3$LE states, reducing the rate of ISC (ref. 7). The solvent also has an influence on the DF behaviour, decreasing the lifetimes from 5.2 to 1.0 μs in 2,8-DPTZ-DBTO$_2$ and 3.0 to 2.6 μs in 3,7-DPTZ-DBTO$_2$. The significant decrease in 2,8-DPTZ-DBTO$_2$ is due to the larger energetic shift. Concomitantly the DF/PF ratio for 2,8-DPTZ-DBTO$_2$ in toluene is reduced to 5 (ref. 13).

**Photophysics in solid state.** The decay profiles, along with the time-resolved emission for each isomer (dispersed in zeonex) are shown in Fig. 9a,b (see Supplementary Table 9 for parameters). For 2,8-DPTZ-DBTO$_2$ the PF has two components, 3.3 and 24 ns, the former is $^1$LE$_D$ emission contribution and the longer dominant contribution is from prompt $^1$CT emission (with very similar lifetime to that measured in MCH). The time-resolved spectra for 3,7-DPTZ-DBTO$_2$, Fig. 9d, shows a clear early time contribution from the axial conformer in contrast to those for 2,8-DPTZ-DBTO$_2$ (Fig. 9c). A $^1$CT$_{ax}$ lifetime of 3.3 ns is accompanied by a prompt $^1$CT$_{eq}$ emission with a lifetime of 9.4 ns, similar to that measured in MCH, this is consistent with $^1$CT$_{ax}$ having much more $^1$LE character than $^1$CT$_{eq}$. Given the similar intensities of both contributions, the observation of the $^1$CT$_{ax}$ emission in the MCH steady-state spectra and the small red-shift of the emission compared to the donor, we ascribe the fast component to the $^1$CT$_{ax}$ state. The viscosity of the surrounding media seems to have little effect on the initial decay. This is in accord with the observation of $^1$CT$_{ax}$ and $^1$CT$_{eq}$ emission in solution, showing that both conformers are stable and do not interconvert in non-polar environments. The delayed fluorescence has a lifetime of 7.4 and 14 μs, for 2,8-DPTZ-DBTO$_2$ and 3,7-DPTZ-DBTO$_2$, respectively. 2,8-DPTZ-DBTO$_2$ again shows more intense DF due to the far larger initial $^3$LE population with a very similar lifetime to that measured in MCH. In

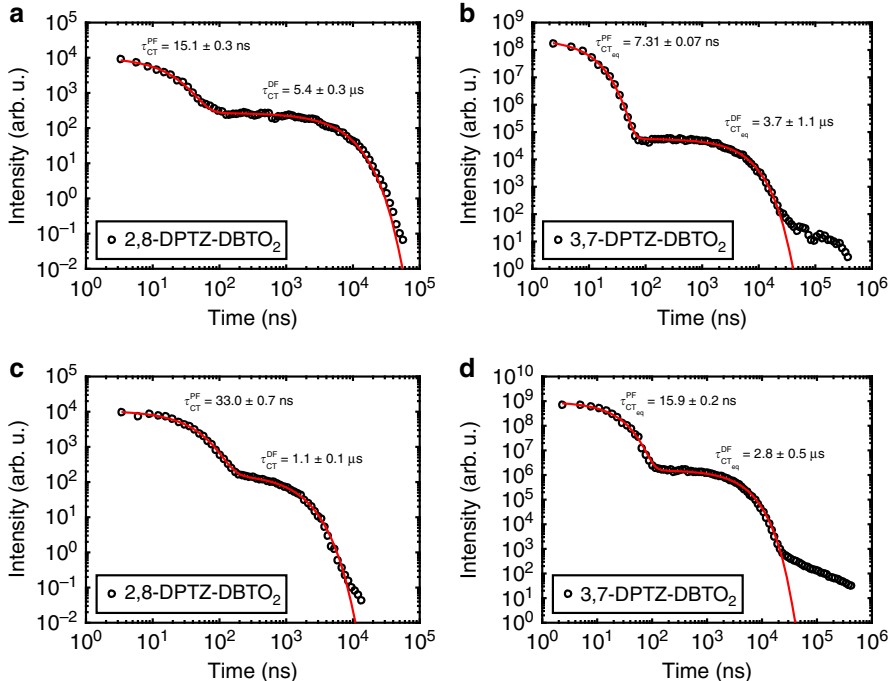

**Figure 7 | Plots of the integrated intensity of the emission of the molecules as a function of time.** (**a**) 2,8-DPTZ-DBTO$_2$ and (**b**) 3,7-DPTZ-DBTO$_2$ in MCH solution and (**c**) 2,8-DPTZ-DBTO$_2$ and (**d**) 3,7-DPTZ-DBTO$_2$ in toluene solution.

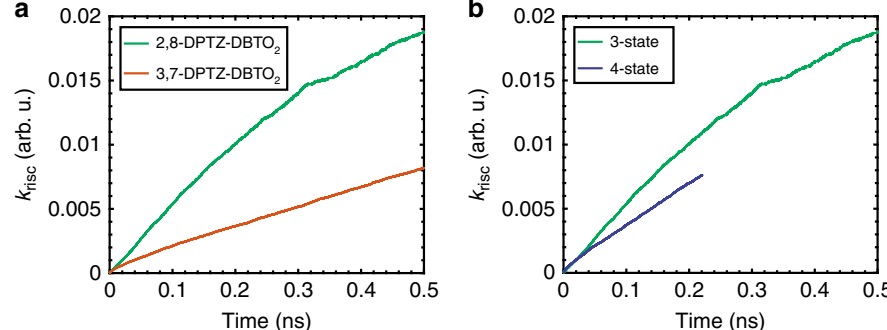

**Figure 8 | Theoretical calculations of the rISC rate as a function of conformer and 3 versus 4 state coupling.** (**a**) The rISC rate in D-A versions of the two isomers, showing that the D-A unit of 2,8-DPTZ-DBTO$_2$ is twice as efficient at rISC compared to the D$_{eq}$-A unit of 3,7-DPTZ-DBTO$_2$. (**b**) Comparing the 3 and 4 state model showing that there is minimal change with the consideration of a fourth state ($^3$LE$_A$). Computational details for these simulations are shown in Supplementary Table 14.

3,7-DPTZ-DBTO$_2$, the DF lifetime is double that of 2,8-DPTZ-DBTO$_2$ indicating very inefficient rISC in keeping with the larger gap. Further, as can be seen in Fig. 9b the DF turns from an exponential into a power law decay. Such a power law decay component is observed in all solid-state samples to varying degrees and we ascribe this to inhomogeneity, that is, molecules with different torsion angles and environments giving rise to an energy dispersion in the CT-$^3$LE gaps which causes a dispersion in rISC rates and thus lifetimes. Finally, phosphorescence is observed with lifetimes in the hundreds of microseconds for both molecules. In this time range the 2,8-DPTZ-DBTO$_2$ spectrum blue-shifts further than 3,7-DPTZ-DBTO$_2$, consistent with the stronger $^3$LE$_A$ contribution. This arises from $^3$LE$_A$ to $^3$LE$_D$ coupling in 2,8-DPTZ-DBTO$_2$, as confirmed by calculations shown in Fig. 8 and the phosphorescence in Fig. 6a. Although the energy gap between the $^3$LE$_A$-$^3$LE$_D$ states means that their coupling is only a minor perturbation to the ISC and rISC rates, it shows that there is a small population of $^3$LE$_A$ arising from the coupling. It is also responsible for the $^3$LE states in 2,8-DPTZ-

DBTO$_2$ exhibiting a single lifetime, despite the $^3$LE$_A$ and $^3$LE$_D$ states having different phosphorescence lifetimes (130 and 64 μs, respectively)[10]. This is not present in 3,7-DPTZ-DBTO$_2$ due to the weaker coupling between the states; a result of their energetic ordering, that is, the low-lying CT$_{eq}$ in zeonex. This type III arrangement and the larger gap between the $^3$LE$_{D,ax}$ and $^3$LE$_A$ means that vibronic coupling is weakened.

We have measured the behaviour of 3,7-DPTZ-DBTO$_2$ in zeonex as a function of temperature (Supplementary Figs 15d and 16d and Supplementary Table 10). Given the rather weak emission, there is no prompt $^1$LE contribution, but emission from both $^1$CT$_{ax}$ and $^1$CT$_{eq}$. The lifetimes of $^1$CT$_{ax}$ is rather insensitive to temperature remaining at *ca.* 4 ns at 80 K. The $^1$CT$_{eq}$ emission lifetime has a weak temperature dependence, however, the DF lifetime stays approximately constant at *ca.* 1.5 μs (within the fitting error) until ∼150 K, when it then lengthens to *ca.* 4 μs at 80 K. This demonstrates weak thermal activation, consistent with a small CT-$^3$LE gap in non-polar zeonex.

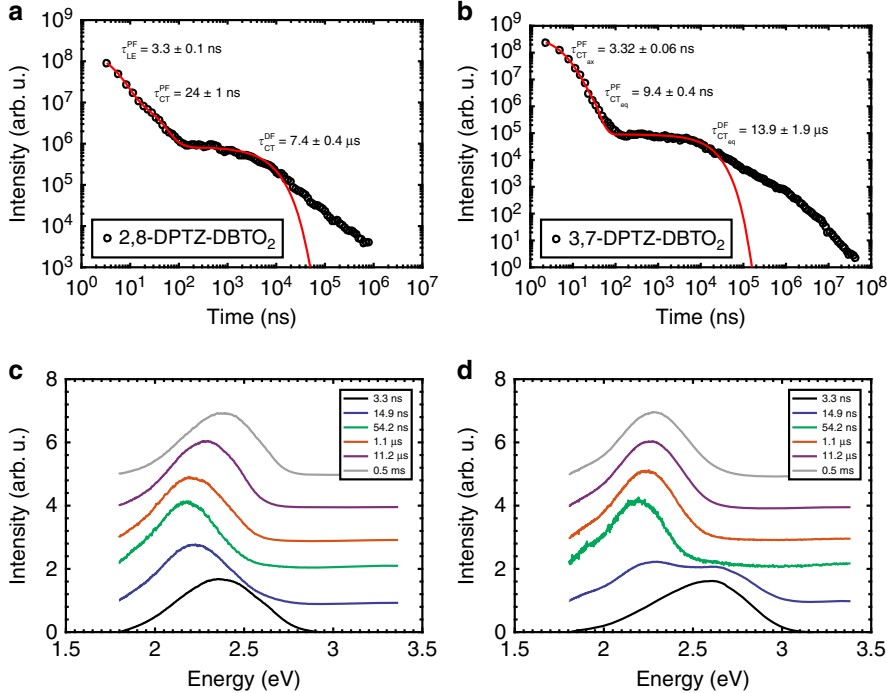

**Figure 9 | Emission decays and time-resolved spectra of the two isomers in zeonex.** (**a**) Emission decay of 2,8-DPTZ-DBTO$_2$ in zeonex. (**b**) Emission decay of 3,7-DPTZ-DBTO$_2$ in zeonex. (**c**) Time-resolved spectra of 2,8-DPTZ-DBTO$_2$ in zeonex. (**d**) Time-resolved spectra of 3,7-DPTZ-DBTO$_2$ in zeonex.

In a dense matrix, CBP (Supplementary Figs 15a and 16a, and Supplementary Table 11), the decay of 2,8-DPTZ-DBTO$_2$ is very simple. A fast decay component with average lifetime *ca.* 5 ns from $^1LE_D$ is observed, which is invariant for temperatures above 160 K. This is accompanied by a longer lifetime prompt $^1CT_{eq}$ component, *ca.* 43 ns, which is constant with decreasing temperature (Supplementary Fig. 15a and Supplementary Table 11). We assume CBP prevents most non-radiative decay apart from ISC, so this behaviour reflects the very small CT -$^3$LE gap in CBP and the down-hill nature of the ISC. The DF component shows a clear monotonic increase in lifetime with decreasing temperature, from 6 μs at 300 K (the same as in zeonex) to 16 μs at 80 K. This is simply due to the decreasing thermal activation and rISC rate, and reflects the up-hill nature of rISC in 2,8-DPTZ-DBTO$_2$. Turning to 3,7-DPTZ-DBTO$_2$ in CBP (Supplementary Figs 15c and 16c and Supplementary Table 12), the prompt decay contains three well-defined exponentially decaying components; we observe *ca.* 1 ns signal corresponding to the ultrafast decay of the $^1LE_D$ state, the $^1CT_{ax}$ component with a temperature invariant lifetime *ca.* 6.5 ns, and $^1CT_{eq}$ decay with lifetime *ca.* 23 ns. The DF component in 3,7-DPTZ-DBTO$_2$ decays faster than in 2,8-DPTZ-DBTO$_2$, a result of the slower rISC and larger radiative rate of $^1CT_{eq}$ decay in 3,7-DPTZ-DBTO$_2$; which is temperature dependent. The lifetime increases from 2.5 μs at 300 K to 14 μs at 80 K reflecting the decreasing thermal energy available to drive the vibronic coupling mechanism. 3,7-DPTZ-DBTO$_2$ in CBP also has a very large power law decay component, which competes with the 'exponentially decaying DF'. This behaviour is strong in CBP because the samples are made by vacuum co-deposition, so 3,7-DPTZ-DBTO$_2$ molecules have significant thermal energy and are rapidly 'frozen' in a particular geometry during deposition. In zeonex, films are deposited from solution so have more time and free volume to attain equilibrium geometry. 3,7-DPTZ-DBTO$_2$ has greater inhomogeneity than 2,8-DPTZ-DBTO$_2$ because of its non-symmetric eq–ax mixed conformer structure. Thus, a large

proportion of the DF will have a longer lifetime and in a device, using a CBP host, we would expect this to manifest itself as increased efficiency roll-off at high drive currents. The time-resolved spectra of 2,8-DPTZ-DBTO$_2$ and 3,7-DPTZ-DBTO$_2$ in CBP at room temperature can be found in Supplementary Fig. 17.

To observe the effect of a changing CT-$^3$LE gap and temperature on the decay trends, we measured 2,8-DPTZ-DBTO$_2$ decays in a polyethylene oxide host, the polarity of which changes with temperature (Supplementary Figs 15b and 16b and Supplementary Table 13). We have shown that DF reaches a resonant maximum at the point when the CT-$^3$LE gap approaches zero, verifying the second-order spin-vibronic mechanism in this system[7]. The fits of the emission decay curves at different temperatures show that the prompt $^1CT_{eq}$ lifetime also follows the resonant behaviour, reaching the longest lifetime of 47 ns at the CT-$^3$LE zero gap point. This can be rationalized because with increasing coupling of $^1$CT-$^3$LE as the gap diminishes, a mixed state evolves where the $^1$CT lifetime lengthens (towards that of $^3$LE) and that of the $^3$LE shortens. At low temperatures, where the gap grows again, the temperature dependence of ISC dominates and the lifetime remains high because of low ISC. The DF follows a similar trend as expected from the theory[9,11] but the lengthening of the DF lifetime is even greater. From these results, we see that the CT-$^3$LE gap dominates ISC and rISC rates whilst temperature has a far smaller effect, especially when the gap is small. For 2,8-DPTZ-DBTO$_2$, in MCH, where there is a small gap, this resonant lifetime increase may also play a part.

**OLED device characterization.** The electroluminescence spectra of the devices made from both isomers are similar to the prompt and delayed fluorescence of the pure emitters (Supplementary Fig. 18b). The maximum external quantum efficiency (EQE) of 2,8-DPTZ-DBTO$_2$ devices was higher (18.2%) compared to 13.3% for 3,7-DPTZ-DBTO$_2$ (Fig. 10). This difference is ascribed

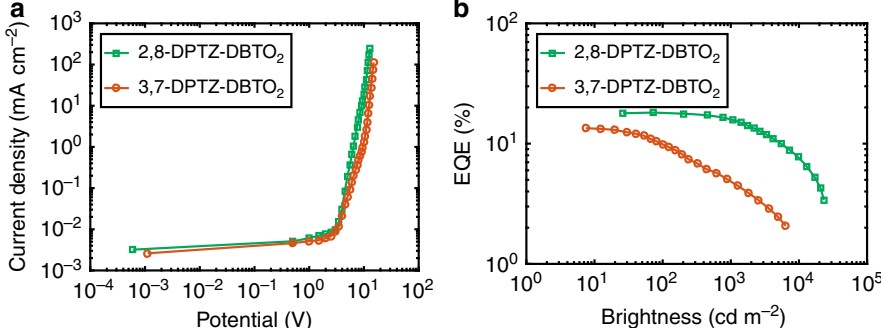

**Figure 10 | The comparison of the 2,8-DPTZ-DBTO₂ and 3,7-DPTZ-DBTO₂ based OLED devices.** (**a**) Current density versus bias, (**b**) EQE versus brightness. The two devices structures are (ITO/NPB (40 nm)/10% 2,8-DPTZ-DBTO₂ in CBP(20 nm)/TPBi (50 nm)/LiF (1 nm)/Al (100 nm)-DEV1; ITO/NPB (40 nm)/ 10% 3,7-DPTZ-DBTO₂ in CBP(20 nm)/TPBi (50 nm)/LiF (1 nm)/Al (100 nm)-DEV2;) A cartoon of these structures is shown in Supplementary Fig. 18a.

to charge-recombination populating both axial and equatorial conformers in 3,7-DPTZ-DBTO₂. Even though the rISC rate is faster and the radiative decay rate of the 3,7-DPTZ-DBTO₂ $^1$CT state is twice that of 2,8-DPTZ-DBTO₂ this loss channel has a marked effect on device efficiency. Given that $^3$LE$_{D,ax}$ effectively quenches the CT$_{ax}$ states, the long residence time in $^3$LE$_{D,ax}$ makes them highly susceptible to polaron excitation quenching in the device[33]. Note, charge-recombination directly creates CT states, not initial $^1$LE$_D$ states[13]. This emphasizes that donor or acceptors that have multiple conformers are problematic for efficient TADF OLEDs.

To further understand the differences that the isomeric structures introduce to device performance, devices of 2,8-DPTZ-DBTO₂ and 3,7-DPTZ-DBTO₂ were also made in a polar DPEPO host[14]. From optical measurements the CT-$^3$LE gap is larger and both systems have slow rISC rates. Again, we find that 2,8-DPTZ-DBTO₂ yields devices, at 7% EQE, that are 1.75 times more efficient than 3,7-DPTZ-DBTO₂ at 4% EQE (Supplementary Fig. 19) concomitant with the reduced rISC efficiency and axial loss channel in 3,7-DPTZ-DBTO₂. Further, we see a stronger efficiency roll-off in the 3,7-DPTZ-DBTO₂ devices through increased polaron excited state quenching.

## Discussion

This study of D–A–D regioisomers has established that the different arrangement of the D and A units between 2,8 (2,8-DPTZ-DBTO₂) and 3,7 (3,7-DPTZ-DBTO₂) stabilizes different conformations of the PTZ units and increases conjugation between donor and acceptor in the 3,7 isomer. In the 3,7 isomer the quasi-equatorial CT, arising from the H-intra PTZ conformer has a lower CT energy that is in resonance with its local triplet state, $^3$LE$_{D,eq}$ and yields efficient TADF. Whereas, the H-extra PTZ conformer is also stabilized in the case of 3,7-DPTZ-DBTO₂. The presence of this mixed conformer structure allows dual CT emission in the system, with the quasi-axial CT state arising from the H-extra PTZ conformer. This state is higher in energy and does not undergo rISC or TADF as a result because of its low-energy $^3$LE$_{D,ax}$ triplet (of this axial CT). This then acts as an effective loss pathway due to this lower-lying 'local axial triplet'. This additional loss channel in 3,7-DPTZ-DBTO₂ is responsible for the reduced device performance compared to 2,8-DPTZ-DBTO₂ because CT excitations are created directly via charge-recombination and so either the quasi-equatorial CT or quasi-axial CT state is populated statistically during charge recombination in a device. This explains why in other D–A–D systems with both PTZ donors in the H-extra conformer no DF at all is observed because of this large S-T gap and low-lying axial triplet

'trap'[27]. In general, a further result of the required vibronic coupling of the $^1$CT and $^3$LE states for efficient rISC and ISC means that when the S–T gap is very small, near zero, the CT and $^3$LE states mix very effectively and the lifetime of the CT state increases through this strong state mixing. In the two isomers studied here, we show how the different conformers are stabilized differently on each isomer yielding different TADF efficiency. Accordingly, we can say that flexible donors and acceptors that exhibit multiple conformers should be avoided in TADF material design[34,35]. Planar donor and acceptor structures are better suited for TADF as they will avoid such conformationally different structures and losses. Further, isomer effects can be used to control the stabilization of different D–A conformers. These new observations and theoretical predictions show how even subtle changes in D–A–D structure radically effect the excited state behaviour and resultant photophysical properties of charge transfer molecules giving a further set of design criteria to be considered.

## Methods

**Optical characterization.** Optical measurements in solution used concentrations in the $10^{-5}$–$10^{-2}$ M range, and samples were deoxygenated using 5 freeze/thaw cycles. (2,8-DPTZ-DBTO₂/3,7-DPTZ-DBTO₂):zeonex films were prepared by spin coating at a ratio of (1:20 w/w). (2,8-DPTZ-DBTO₂/3,7-DPTZ-DBTO₂)/CBP films were prepared by co-evaporation (10% weight of dopant). Absorption and emission spectra were collected using a UV-3,600 double beam spectrophotometer (Shimadzu), and a Fluorolog fluorescence spectrometer (Jobin Yvon).

**Time-resolved emission decay.** Phosphorescence, prompt fluorescence (PF), and delayed emission (DF) spectra and decays were recorded using nanosecond gated luminescence and lifetime measurements (from 400 ps to 1 s) using either a high-energy pulsed Nd:YAG laser emitting at 355 nm (EKSPLA) or a N₂ laser emitting at 337 nm. Emission was focused onto a spectrograph and detected on a sensitive gated iCCD camera (Stanford Computer Optics) having sub-nanosecond resolution[32].

**Photoinduced absorption.** The quasi-CW PIA measurements of the excited state absorption (and emission) spectra[28], were performed using a 375 nm pump beam (Vortran Stradus 375-60) modulated at 73 Hz, with a continuous laser driven white light source (Energetiq EQ-99X) as the probe. The probe beam was then passed through a Bentham TM300 monochromator and incident on a Si detector connected to the Signal Recovery dual channel 7,225 digital lock-in amplifier that also provides the reference frequency modulation for the pump laser.

**Photoluminescence quantum yield.** The PLQY measurements were performed using a Quantaurus-QY Absolute PL quantum yield spectrometer. Initially a background was taken using a clear substrate and then the (2,8-DPTZ-DBTO2/3,7-DPTZ-DBTO2):zeonex films were measured both under N2 and in air.

**Device fabrication.** All organic evaporated compounds were purified by vacuum sublimation using a Creaphys organic sublimation system. The suppliers and the full chemical names of the materials used are as follows: CBP—4,4′-bis

(N-carbazolyl)-1,1′-biphenyl (Sigma Aldrich), NPB—N,N′-di-1-naphthyl-N, N′-diphenylbenzidine (TCI-Europe), TPBi—2,2′,2″(1,3,5-benzenetriyl)-tris (1-phenyl-1H-benzimidazole) (LUMTEC), LiF (99.995%, Sigma Aldrich) and aluminium wire (99.9995%, Alfa Aesar). OLED devices were fabricated using pre-cleaned indium-tin-oxide (ITO) coated glass substrates purchased from Ossila with a sheet resistance of $20\,\Omega\,cm^{-2}$ and ITO thickness of 100 nm. The OLED devices had a pixel size of 2 mm by 1.5 mm. The small molecule and cathode layers were thermally evaporated using a Kurt J. Lesker Spectros II evaporation system, and the deposition pressure was $10^{-6}$ mbar. All organic materials and aluminium were deposited at a rate of $1\,\text{Å}\,s^{-1}$ with the LiF layer deposited at $0.1\,\text{Å}\,s^{-1}$. The IV characteristics of the OLED devices were measured in a 10-inch integrating sphere (Labsphere) connected to a Source Meter Unit.

**Quantum chemistry.** Quantum dynamics simulations probing the mechanism for efficient rISC were performed using the density operator formalism of the multi-configurational time dependent Hartree method[36]. Here we adopt a closed quantum system, using the Hamiltonian described in ref. 6. The energy gap between the $^1CT$–$^3CT$ states was as calculated in ref. 20, however the gap between the $^3LE$–$^3CT$ states was extracted experimentally. The full details of the simulations and the model Hamiltonians used is provided in the Supplementary Information.

**Data availability.** Data supporting this publication are openly available under an 'Open Data Commons Open Database License'. Additional meta-data are available at: 10.17634/153015-2. (theoretical) and 10.15128/r2df65v784t (experimental). Please contact Newcastle Research Data Service at rdm@ncl.ac.uk for access instructions.

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

## Acknowledgements

A.P.M. acknowledges the EU's Horizon 2020 for funding the PHEBE project under grant no. 641725 and the EPSRC for funding under grant number EP/L02621X/1. T.J.P. acknowledges the EPSRC, Project EP/N028511/1 for funding. P.D. thanks the EU for a Marie Curie Fellowship H2020 research and innovation programme under grant agreement no. 691684. A.K. thanks networking action funded from the European Union's Horizon 2020 research and innovation programme under grant agreement no 659288. P.L.D.S. thanks CAPES Foundation, Ministry of Education of Brazil, Science Without Borders Program for a PhD studentship, Proc. 12,027/13-8. F.F. thanks CAPES Foundation, Ministry of Education of Brazil, Science Without Borders Program for a Fellowship.

## Author contributions

M.K.E., F.B.D. and F.F. performed most of the optical measurements. The full physical model was devised by M.K.E., J.G., T.N., T.J.P. and A.P.M. The simulations were performed by J.G., T.N. and T.J.P; J.S. and J.S.W. synthesized and characterized the molecules under the supervision of M.R.B.; H.F.H. performed the high concentration absorption studies. P.D. conducted the cyclic voltammetry, fabricated and measured devices in CBP host. A.K. made the ultraviolet–visible and EPR measurements. P.L.D.S. fabricated and measured the devices in a DPEPO host. D.R.G. measured the time-resolved emission of the samples in a CBP host. A.S.B. solved the X-ray crystal structures. A.P.M. and M.R.B. conceived the original idea of isomeric structures for investigation.

## Additional information

**Competing interests:** The authors declare no competing financial interests.

