## [Peer Review File · Nature Communications]

Reviewers' comments:

Reviewer #1 (Remarks to the Author):

In the present manuscript, the authors present a series of D-A-D compounds that allows for exploration of isomerization structural parameter that influence the TADF behavior. The topic is valuable and the work is well presented. The arguments of the content is supported well with appropriate studies (i.e. photo physical characterization, CV, ultrafast transient absorption, etc.). I think this manuscript is suited for publication in Nature Communications after minor revision. Below, I have some minor points which I would like to offer the authors for consideration to improve the manuscript.

1. In Figure 2, the authors demonstrated the solvatochromism of CT states emission in 3,7-DPTZ-DBTO2 and the energy levels of quasi-equatorial CT and quasi-axis CT states is obtained. Since the absorption spectra is complicated and superpositioned, the authors should provide more information about handling data to obtain the actual energetic value.
2. Solvatochromism experiments have been performed and varieties of solvent with different polarity were used. What the abbre of "MCH" for? Methylcyclohexane? Or something else?
3. How the authors excluded the possibility of energy transfer from 3LEDeq to 3LEDax?
3. Some typo and spelling errors should be corrected. such as Page 5, line 16 "not" to "note"

Reviewer #2 (Remarks to the Author):

Authors report on the study of two regioisomers exhibiting TADF. They study with great details the photophysics of intersystem crossing from the CT state and reverse intersystem crossing back to the luminescent CT state using time resolved spectroscopy. This allows them to conclude that conformational degrees of freedom provide additional relaxation channels which have negative repercussions for TADF. They investigate the performance of efficient OLEDs built on these compounds and show how time resolved spectroscopy correlates with the EQE. The study is substantiated by TDDFT calculations.

This represents a very significant amount of data and data analysis.

Paper is well written, rather concise and very convincing.

It represent an advance in the understanding of the TADF effect for efficient OLEDs, mainly the reverse intersystem crossing process and how it correlates with internal degree of freedom.

I recommend publication in the present state.

Reviewer #3 (Remarks to the Author):

In this work, dibenzo[b,d]thiophene-S,S-dioxide was substituted with a donor moiety at 2,8- position or 3,7- position to study the effect of the regioisomerization on the excited state and TADF properties of thermally activated delayed fluorescent emitters. 2,8- position substituted compound showed one quasi-equatorial conformer on both donor sites and charge-transfer (CT) emission close to the local triplet state. Eventually, the 2,8- position substituted material showed efficient TADF. On the other hand, 3,7- substituted compound, showed a quasi-equatorial CT state and a higher-energy quasi-axial CT state and no TADF was observed. In fact, the positional effect of the donors was already reported in other publications. The same author also reported the same result a few years ago, but this work covered detailed points about the molecular structure.

Strong point of this work is that the authors clarified the origin of the TADF behavior of the 2,8- position substituted material by analyzing the photophysics of the materials.

Weak point of this work is that the proposal of this work is specific in some donor moiety rather than

general applicability. Only D-A-D structure is within the category proposed in this work among many designs reported in the literature.

To strengthen this work, other donor moieties should also be considered except for phenothiazine, phenoxazine or acridan. This work can be more important if general conclusions can be drawn from other molecules with carbazole type donors.

REVIEWERS' COMMENTS:

Reviewer #1 (Remarks to the Author):

I'm satisfied with the response from the authors. I recommend acceptance of the manuscript for publication in Nature Communications.

Reviewer #3 (Remarks to the Author):

Authors provided reasonable answers and revised version of the manuscript according to reviewer's comments. I acknowledge that this work is ready for publication.

Response to Reviewers

We would like to thank the referees for their constructive comments and we hope the changes made to the manuscript and our comments below will satisfy their queries.

Reviewer 1

- 1) We are grateful for the reviewer highlighting, and have taken the opportunity to explain, our data handling of the CT state energetics, specifically in the 3,7 isomer. In Figure 2b, the arrows and energies referred to the peak of the quasi-axial CT emission mainly as a guide to the reader to identify the solvatochromism. However, these have now been removed to avoid confusion as the onset energy is the more important and extra information has been provided in the caption to explain the estimation of these onset energies, which are the important energies for the calculations and modelling. The method for measuring onset energies can be found in our recent communication¹, however Supplementary Figure 25 has now been added as an example. Extracted energies for Figure 2b can now be found in Supplementary Table 15. These differ slightly to those in Figure 1c as those values have been interpreted from a range of different spectra e.g. time-resolved, etc. to ensure greater accuracy.
- 2) MCH does indeed stand for methylcyclohexane and changes have been made to the manuscript to reflect this.
- 3) The reviewer wonders if there is possible energy transfer from the higher energy $^3\text{LE}_{\text{D,eq}}$ to the lower energy $^3\text{LE}_{\text{D,ax}}$ in the 3,7 isomer. We have not looked directly at this but point out that in the 3,7 isomer the axial and equatorial conformation donor units are orthogonal to each other which usually strongly inhibits energy transfer, further we observed photoinduced absorption signals characteristic of both triplets in the 3,7 isomer showing their co-existence. Given also that the singlet states of the two conformers coexist then we feel that there is little to support a proposal for energy transfer between the conformers. This is an interesting aside and we will give thought to how it might be resolved in future work.
- 4) The paper has again been rechecked for typographical errors.

Reviewer 2

We thank Reviewer 2 for their extremely positive responses.

Reviewer 3

We thank Reviewer 3 for their comment summary and acknowledgement of the strength of this work. We hope that our comments below and highlighted changes in the manuscript address the observed weaknesses.

While the study is particular to the phenothiazine donor, the crucial identifier of most of the phenomenon discussed here arise from the conformational behaviour of the donor moieties and their bonding position. There have been noted studies on the fact that phenothiazine (itself) can form two conformers, and these authors and further work also indicate that there are other 'potential' donors that also display such molecular conformations due to their flexibility of structure (isoalloxazine, acridan, xanthene, thioxanthene)^{2,3}. (These further references detailing examples of non-planar materials that may behave like phenothiazine have been added to the main text in the conclusion to emphasise this). There are also donor molecules that are rigid and do not exhibit these conformers e.g. carbazole, phenoxazine and acridene, and we point out that these are unable to form two CT states in D-A-D molecules. We have given x-ray crystal structures for the phenoxazine D-A-D analogue in recent work⁴ for example. Thus, from a purely conformational standpoint rigid molecules like carbazole will be a better choice than flexible molecules like phenothiazine (stated in the manuscript). However, other parameters like their donating strength, emission wavelength and will still need to be considered. Thus, we feel that while focusing on phenothiazine (as an archetypical flexible donor) to highlight the effect of donor and hence D-A-D conformational effects, it will be clear to the reader what causes these effects and thus, generally, which donors to be aware of that can also give rise to conformational isomers and potential efficiency quenching in a TADF OLED.

1. Etherington, M. K., Gibson, J., Higginbotham, H. F., Penfold, T. J. & Monkman, A. P. Revealing the spin–vibronic coupling mechanism of thermally activated delayed fluorescence. *Nat. Commun.* **7**, 13680 (2016).
2. Malrieu, J.-P. & Pullman, B. Configuration spatiale et proprietes electroniques du noyau d'isoalloxazine. *Theor. Chim. Acta* **2**, 302–314 (1964).
3. Aizenshtat, Z., Klein, E., Weiler-Feilchenfeld, H. & Bergmann, E. D. Conformational Studies on Xanthene, Thioxanthene and Acridan. *Isr. J. Chem.* **10**, 753–763 (1972).
4. dos Santos, P. L., Ward, J. S., Bryce, M. R. & Monkman, A. P. Using Guest-Host Interactions to Optimize the Efficiency of TADF OLEDs. *J. Phys. Chem. Lett.* **7**, 3341–3346 (2016).